# A Review of Silver Wire Bonding Techniques

**DOI:** 10.3390/mi14112129

**Published:** 2023-11-20

**Authors:** Bin An, Hongliang Zhou, Jun Cao, Pingmei Ming, John Persic, Jingguang Yao, Andong Chang

**Affiliations:** 1School of Mechanical and Power Engineering, Henan Polytechnic University, Jiaozuo 454003, China; a1951198730@163.com (B.A.); mpm@hpu.edu.cn (P.M.); yaojingguang1205@163.com (J.Y.); cadzmy@163.com (A.C.); 2Henan International Joint Laboratory of Advanced Electronic Packaging Materials Precision Forming, Henan Polytechnic University, Jiaozuo 454003, China; 3Microbonds Inc., 7495 Birchmount Rd., Markham, ON L3R 5G2, Canada; jpersic@microbonds.com

**Keywords:** silver wire bonding, manufacturing process, bonding parameters, bonding reliability, development trends

## Abstract

The replacement of gold bonding wire with silver bonding wire can significantly reduce the cost of wire bonding. This paper provides a comprehensive overview of silver wire bonding technology. Firstly, it introduces various types of silver-based bonding wire currently being studied by researchers, including pure silver wire, alloy silver wire, and coated silver wire, and describes their respective characteristics and development statuses. Secondly, the development of silver-based bonding wire in manufacturing and bonding processes is analyzed, including common silver wire manufacturing processes and their impact on silver wire performance, as well as the impact of bonding parameters on silver wire bonding quality and reliability. Subsequently, the reliability of silver wire bonding is discussed, with a focus on analyzing the effects of corrosion, electromigration, and intermetallic compounds on bonding reliability, including the causes and forms of chlorination and sulfurization, the mechanism and path of electromigration, the formation and evolution of intermetallic compounds, and evaluating their impact on bonding strength and reliability. Finally, the development status of silver wire bonding technology is summarized and future research directions for silver wire are proposed.

## 1. Introduction

Wire bonding technology is one of the most critical technologies in the field of semiconductor packaging, which utilizes mechanical energy, ultrasonic energy and thermal energy to connect chips to solder pads on packaging substrates using metal wire [1,2]. Since the 1990s, gold wire bonding has been dominant. Compared with other bonding materials, gold wire bonding has excellent mechanical strength and high reliability. But as the price of gold continues to rise, finding alternatives to gold wire has become a hot topic [3,4]. Copper wire has better mechanical and electrical properties and is low-cost, replacing gold wire on a large scale in the past period of time. However, the intermetallic compounds formed at the Cu/Al interface often fail to ensure good bonding of electronic packaging. More troubling is the inherent oxidation tendency of copper, which has raised serious concerns about the reliability of electronic products. Such oxidation and corrosion problems cannot be completely avoided, even if precious metals such as palladium, gold or platinum are used to coat welding wires. In addition, compared to other wire materials, its hardness is higher, which can cause damage to the bonding pad during the bonding process. Therefore, copper wire bonding has many limitations [5]. In many fields, especially in the automotive and military fields, the vast majority of enterprises will not choose to replace gold wire with copper wire. Silver wire bonding is a new interconnecting method in semiconductor packaging. Silver has the highest thermal conductivity and electrical conductivity among silver, copper and gold. Silver has an excellent ability to bond with various different pad materials to form low-resistance primary interconnects [6]. The hardness of pure silver is between that of copper and gold, similar to gold, and the cost of silver wire is about 1/20 that of gold wire. Silver wire is also superior to gold and copper wire for use in intermetallic medium micro-cracks. The researchers believe that the moderate rate of Ag-Al IMC formation and growth and the ease of mass production are key factors for the success of Ag wire bonding instead of Au or Cu wire. Taking into account the above advantages of silver, more and more researchers are attempting to replace gold wire with silver wire. However, pure silver has issues such as rapid degradation of IMC after thermal aging, susceptibility to corrosion in humid environments, and electromigration, making it unsuitable as a bonding wire [7,8]. In order to solve this dilemma, researchers have begun research on alloy silver wire, coated silver wire and other wire materials, in order to find solutions to the defects of pure silver wire. This paper reviews the research progress of silver bonding wire, as well as the manufacturing and bonding processes of silver bonding wire, and discusses the reliability of silver wire bonding.

## 2. Types and Characteristics of Silver Bonding Wire

### 2.1. Pure Silver Wire

Pure Ag has excellent electrical heating properties, similar to gold bonding wire, resistance to deformation, a large bonding parameter window and strong bonding adaptability. In addition, silver wire also has the characteristics of easy ball formation. Compared with copper wire, silver wire has a higher oxidation and vulcanization resistance. Table 1 lists the key properties of commonly used pure-element bonding wire materials [9,10,11]. Hsueh et al. [12] loaded direct current into 23 μm silver bonding wire via a current experiment, observed the structural changes and interfacial characteristics of Ag–Al bonding, and the experimental results showed that the silver wire and aluminum pad had relatively excellent bonding characteristics. After observing the many advantages of silver wire, more and more researchers began to study silver bonding wire.

However, the drawbacks of pure silver wire are also evident. Under high-temperature conditions, pure silver wire bonding is prone to neck fracture, as shown in Figure 1, which affects its service life. The long-term use of pure silver wire is prone to sulfide and chlorination phenomena, as shown in Figure 2, resulting in low bonding reliability [13]. There are many sulfur-containing substances in nature that produce sulfur molecules. Especially in an environment of high temperature and high light radiation, silver easily reacts with sulfur molecules, resulting in the formation of silver sulfide and corrosion [14]. Hsieh et al. [15] found via experiments that a Ag atom could migrate from a wedge bond to a spherical bond via the wire surface under the action of three driving forces: diffusion, alloy formation and galvanic effect. Hassan et al. [16] studied the electrochemical behavior of silver in NaCl, NaBr and NaI aqueous solutions, and pointed out that porous halide films would be formed when silver was dissolved in halide solutions. Ha and Payer [17] studied the dissolution behavior of silver in a NaCl solution. They found that silver corrodes and grows into a AgCl film in a solution containing chloride. Ag ions move through the gaps between AgCl membranes. As the AgCl film thickens and as these spaces close, silver continues to dissolve through the microchannels formed in the AgCl film. In electronic packaging, polymer packaging agents typically contain corrosive chloride ions and have hygroscopicity, which can absorb water from the environment and cause corrosion of silver bonding wire. In addition, pure silver is prone to oxidation, resulting in low IMC coverage, and the FAB neck strength of pure silver wire ball bonding is low. There is a high possibility of neck cracks and failure under high-temperature conditions.

### 2.2. Alloy Silver Wire

Compared with gold bonding wire, copper bonding wire and pure silver bonding wire, silver-based bonding wire containing Au and Pd is attracting more and more attention due to its higher tensile strength and higher anti-vulcanization and anti-oxidation properties [18]. In order to solve the various defects of pure silver wire, researchers began to attempt to add elements such as Au, Pd, Pt, Ru, Zn, etc. to pure silver to make alloy silver bonding wire. Doping different elements has different effects on material properties. For example, doping Pd and Au can improve the mechanical properties of silver alloy materials; doping Sn can reduce the melting point of silver alloy materials, improve the feasibility of the process and the plasticity of bonding materials, and reduce the melting point and the cost of the preparation process; doping In can improve the ductility of bonding materials and help improve the stability of bonding processes.

Fan et al. [19] studied bonded Ag-Au alloy wire with different Au contents. The experimental results show that, with an increase in Au content, the melting point of the Ag-based bonding wire increases, and the sphericity of FAB becomes better. HAZ decreases and bond strength increases. The tension and shear force of the free air ball will be increased, and the bonding reliability will be improved. The study by Yuan et al. [20] also confirms that the addition of palladium can improve the mechanical properties of silver wire. The initial tensile strength of the several alloy wires they tested was higher than that of pure gold wire, indicating that the thermal aging sustainability of alloy wire was at least as good as that of pure gold wire. After adding Pd to the silver-based bonding wire and bonding with the Al substrate, a palladium oxide protective layer is formed, resulting in better performance in bond shear testing (BST). Kuo et al. [21] found that, although the hardness and tensile strength of silver-based alloy wire would be increased with an increase in Au content, its elongation and conductivity would be decreased.

The addition of Au and Pd to Ag can also effectively reduce the migration of Ag ions. Chuang et al. [22] evaluated the ion migration resistance in silver alloy wire using water drop tests and found that, with an increase in the content of palladium and gold, the silver ion migration rates of various binary silver–palladium and ternary silver–palladium alloy wires showed a downward trend, and the inhibiting effect of the Pb element on silver ion migration was better than that of the Au element. The relationship between Ag mobility and Pb and Au content is shown in Figure 3. Cho et al. [23] found via experimental studies that interfacial corrosion was significantly inhibited with an increase in palladium content, and the reliability of Ag wire bonding containing palladium was significantly improved. Chuang et al. [24] found that alloys with higher Pd content were conducive to the formation of Ag_2_Al intermetallic compounds (IMC), while alloys with lower Pd content caused serious cracking due to the formation of Ag_3_Al intermetallic compounds (IMC). In their research, they observe that the bonding interface between Ag-2Pd, Ag-10Pd and Ag-12Pd silver alloy wire and an Al substrate is relatively good, as shown in Figure 4, with high bonding strength that has reached the industrial standard. In addition, their study confirmed the immobility of Pd and suggested that doping of Pd should be limited to a certain range, as insufficient diffusion of Pd clearly led to the formation of cracks at the bonding interface. Silver is easily oxidized, and the addition of palladium and gold can also improve the oxidation resistance of silver-based bonding wire. Cheng et al. [25] produced a kind of Ag-Au-Pd alloy wire with excellent antioxidant properties, which passed various harsh environmental tests and high-standard production line tests, and that was successfully applied to batch production lines. Tsai et al. [26] mentions that ternary Ag-8Au-3Pd wire is expected to become an ideal substitute for traditional Au wire due to its high strength, corrosion resistance and reliability, while binary Ag-3Pd and Ag-4Pd wire is expected to become a cost-friendly alternative bond wire for high-frequency integrated circuit devices.

At present, palladium and gold are the most-studied elements doped into silver-based bonding wires, and the related studies of other elements doped into silver to make silver alloy wire are relatively few. Cao et al. [27] added Zn to a silver alloy and found that Zn reduced the thermal conductivity of the bonding wire and avoided the formation of golf balls and sharp balls. In addition, the addition of the Zn element can also reduce the migration rate of silver atoms and avoid cracks at the bonding interface. By doping alloy elements such as cerium and lanthanum, the corrosion resistance and oxidation resistance of silver alloy wire can be improved, the growth and diffusion rate of IMC can be reduced, and the reliability of silver alloy wire bonding points can be enhanced [28]. Hsueh et al. [29] doped lanthanum into a silver matrix for the first time to make Ag-La alloy wire. They found that the addition of lanthanum improved the oxidation resistance of the silver-based bonding wire, reduced the diameter of the FAB and strengthened the hardness of the matrix. Cao et al. [30] studied a Ag-11Au-4.5Pd-0.1Y alloy bonding wire and found that yttrium could reduce the migration rate of silver. In addition, the length of the heat-affected zone is reduced by 30%, and the neck strength is sufficient, so the neck will not crack, and the cracking of the bonding interface is largely avoided. These pioneering studies on the bonding wires of doped rare earth elements provide an important reference for the application of rare earth elements in the field of lead bonding.

### 2.3. Coated Silver Wire

Coated silver wire is a metal film coated onto the surface of silver-based bonding wire to improve its performance. Gold and palladium are currently the most-studied silver-based bonding wire coating materials.

Gold-coated silver wire is also known as ACA wire. Via the study of ACA wires, Tseng et al. [31] found that ACA wires had excellent electrical properties. The coating of a layer of gold onto the surface of silver wire can protect the surface from oxidation, improve its oxidation resistance, and also enhance its tensile strength. Gold-coated silver wire can improve the bonding strength and surface properties of silver wire, and is a promising material for development. Zhou et al. [32] studied gold-coated silver alloy (ACAA) wires with Pd contents of 0 wt%, 1 wt% and 4 wt%, respectively (labeled as wires I, II and III). ACAA wires were prepared by physically coating a Au layer of the same thickness. They found that, with an increase in Pd content, the tensile strength, elongation, Young’s modulus, hardness, and resistivity of the gold-coated silver wire all showed an upward trend. Hong et al. [33] found that the corrosion resistance of gold-coated silver alloy wire was better than that of pure silver wire and slightly lower than that of gold wire. They also studied the corrosion potential (ECORR) values of ACA wire in different electrolytes and suggested that current and voltage ranges should be considered in the use of ACA wire. The cross-sectional structure of ACA core material studied by Kang et al. [34] is an equiaxed structure with small grain size and a large number of lath annealing twins. In terms of mechanical properties, electrical properties and bonding properties, the test results are better than those of gold wire and pure silver wire. Kim et al. [35] found in their study that there was a thin Ag-Au-Al intermetallic compound layer at the ACA binding interface which prevented crack propagation, as shown in Figure 5. Moreover, Ag-Au-Al intermetallic compounds formed at the bonding interface of ACA wire hinder the Au-Al interdiffusion reaction and reduce the intermetallic diffusion. Cao et al. [36] studied gold-coated silver wire with different coating thicknesses and found that a too-thin coating would lead to FAB deflection and an unstable shape of ball solder joints in the EFO process. An excessively thick coating can lead to FAB sharpening. Therefore, the coating is not the thicker the better and it should be limited to a certain range.

Palladium coating on silver wire is similar to palladium coating on copper wire, which can change the mechanical and electrical properties of the material. At present, the related research on palladium-coated silver wire is still relatively scarce, and the space for further research is relatively large. Tana et al. [37] prepared a palladium-coated silver alloy wire (PCS) that exhibited better FAB diameter consistency and superior mechanical properties compared to gold wire. Mayer et al. [38] successfully achieved the ball welding of PCS bonding wire without using any protective gas. Compared with the control group of Au ball bonds, PCS ball bonds showed a significant increase in resistance during high-temperature aging, but their lifespan was longer without complete failure. This indicates that the Pd coating on silver wire has a beneficial impact on bonding reliability. Patent [39] discloses a composite-coated silver alloy wire, with the coating sequence being pre-gold-coated, palladium-coated and double-gold-coated. This material has good plastic deformation ability and low cost.

In addition to the common Au and Pb coatings, there are also related studies on Pt coatings. Patent [40] has invented a platinum–silver-coated alloy wire that can meet the requirements of thrust, tension, mechanical strength and other aspects in the field of wire bonding. It has better conductivity, thermal conductivity and reliability than traditional pure alloy wire, and has advantages such as an optimized arc height and a low breaking rate. The research on coated silver wire is still in its initial stage, and related research still needs to be strengthened. The bonding wire coating process mostly uses strong acids, alkalis and organic solutions, and improper treatment can cause environmental pollution and other problems. Therefore, there are still many issues to consider in the development and preparation of coated silver bonding wire. Therefore, from preparation to mass production, more manpower and resources need to be invested into the further development and research of replacing gold wire with coated silver bonding wire.

## 3. Silver Wire Manufacturing and Bonding Technology Development

### 3.1. Silver Bonding Wire Manufacturing Process

Similar to other bonding wires, the research on the processing process of silver bonding wire can pave the way for the mass production of silver bonding wire, which is of great significance for the large-scale replacement of gold bonding wire by silver bonding wire. Annealing and wire drawing are the key processes in the production of bonded wire. The research on them can find out the problems in the production processes of different bonded wires and provide theoretical support for the improvement of the manufacturing process of bonded wire.

Annealing is a common processing technology for bonding wire. Annealing can change the mechanical properties of materials and has a broad application prospect in the field of electronic packaging. Cao et al. [41] studied the effect of heat treatment temperature on Ag-4Pd alloy wire and found that, with an increase in annealing temperature, the tensile strength of Ag-4Pd alloy wire decreased, the elongation increased and the resistance value decreased and then tended to be stable. They also studied the effects of cold deformation processing rate and annealing temperature on the mechanical properties, microstructure and melting current of Ag-4Pd bonded alloy wires. The results show that the strength of the bonded alloy wire increases and the elongation decreases with an increase in cold working rate. The bonding alloy wire has excellent mechanical properties when heat-treated at 525 °C. In addition, it is also found via the experimental data that there is an exponential function relationship between the fuse current and the fuse time of the wire, and a polynomial function relationship between the fuse current and the arc length of the wire [42]. Tseng et al. [43] studied the effect of annealing on another Ag-2Pd alloy wire and found the same rule, that is, as the annealing temperature increased, the tensile strength of the alloy wire decreased and the elongation increased. Hsueh et al. [44] studied annealed pure silver wire and concluded that the effect of annealing on the performance of pure silver wire was consistent with the above law. This shows that annealing can reduce the tensile strength of silver bonded wire and increase the elongation of silver bonded wire. This law is generally applicable to silver bonded wire. In addition, annealing can also improve the electrical properties of bonded wires. The change in metal resistivity is mainly caused by phonons, dislocations, point defects (soluble atoms, impurities, vacancies, etc.) and the scattering effect of the interface on electrons [45]. Cao et al. [46] found that annealing would lead to the drastic migration of point defects and dislocations, which would affect the microstructure of materials and lead to a change in material resistivity. Kong et al. [47] found via an annealing experiment of Cu-Ag alloy bonding wire that the conductivity of bonding wire increased with an increase in annealing temperature.

Annealing twin is beneficial for maintaining the thermal stability of grain structure and mechanical properties at high temperatures of the bonding wire, and is conducive to the development of high-reliability bonding wire materials. Chuang et al. [48] produced an innovative Ag-8Au-3Pd bonding wire containing a large number of annealing twins via appropriate stretching and annealing processes. In the experiment, they observed that the grain size of the Ag-8Au-3Pd wire with annealing twins almost remained unchanged at different times, while obvious grain growth could be observed in the conventional Ag-8Au-3Pd wire. The anti-electromigration durability of the annealed-twin Ag-8Au-3Pd alloy wire was also higher than that of the conventional-grain wire [49]. The bonding wire also has a significant advantage, that is, with the extension of aging time, its tensile strength and elongation increase at the same time. Chen et al. [50] used EBSD analysis to study the annealing treatment of Ag-4Pd ribbon. Figure 6a–c show the SEM image of the Ag-4Pd ribbon material after annealing for 6 s at 873 K, the reverse pole diagram of crystal orientation (IPF) and the characteristic grain boundaries of the prepared Ag-4Pd ribbon, respectively. It was shown that the material underwent recrystallization at 873 K and to some extent completed recrystallization, even in a short period of 6 s. They also used EBSD to simulate the equivalent grain diameter with grain orientation to obtain grain size. The prepared Ag-4Pd ribbons were annealed for 100 h at annealing temperatures of 623 K, 773 K, 923 K and 1073 K, respectively, and their grain growth was observed. The microstructure is shown in Figure 7. The results show that the microstructure is almost unchanged when the annealing temperature is lower than the manufacturing temperature, but the grain growth is obvious when the annealing temperature is higher.

Cold rolling and wire drawing processes also have a great influence on the properties of materials. Xie et al. [51] found via their research on pure copper and Cu–Ag alloys that cold rolling could change the strength and hardness of materials. In the process of cold rolling, obvious strain hardening occurred in all materials. In the initial strain state, the hardness of materials increased with an increase in rolling strain, as shown in Figure 8. However, the hardness finally reaches a stable state under large strain rolling, and the improvement in strength and hardness during cold rolling is mainly controlled by dislocation strengthening and boundary strengthening. Hou et al. [52] studied the preparation of platinum–silver alloy ultrafine wire, proposed a reasonable drawing process for platinum–silver alloy wire, and successfully drew the wire to 0.02 mm. Zhu et al. [53] proposed a new type of continuous casting and continuous drawing processing method for the preparation of micro-wires, which had the advantages of short process and high efficiency, providing a new idea for the bonding wire manufacturing process.

After noting the various excellent performances of coated silver-based bonding wire, research on the preparation process of coated silver wire is also constantly updated and developed. Li et al. [54] mentioned that cyanide gold coating had always been the mainstream process in the gold-coating industry. Due to the characteristics of stable coating solution, high brightness of coating, strong bonding ability, high flatness and strong dispersion ability, it is widely used in the packaging materials of medium- and high-end electronic components. However, cyanide gilding is highly toxic, which not only affects the health of the operator, but also seriously contaminates the environment with high risk. Therefore, a cyanide-free gold-coating process is gradually replacing cyanide gold-coating process, and the electrical and mechanical properties of gold-coated silver bonding wire produced using the cyanide-free process are better. In addition to the coating process being constantly updated and iterated, the alloying manufacturing process of silver-based bonding wire is also constantly optimized. The patent [55] discloses a preparation method for palladium alloy reinforced composite bond wire. Unlike conventional methods for preparing palladium–silver alloy wire, this method utilizes liquid-phase chemical synthesis and powder metallurgy techniques to avoid the problem of increased electrical resistance caused by the formation of Ag alloys with elements such as Ag and Pd. As a result, the silver–palladium alloy prepared using this method exhibits superior electrical conductivity compared to that prepared using traditional methods. Furthermore, material costs can be reduced by over 60% compared to those of bonded alloy wire.

In recent years, in addition to conventional metal coating, various new surface treatment techniques have continuously emerged. The chemical activity of the metal surface can be reduced by passivating the metal surface with a corresponding chemical reaction. Zhou et al. [56] proposed a passivation process in which a dense, uniform and continuous transparent film with a thickness of 3 nm was coated onto the surface of the passivated silver alloy welding wire, as shown in Figure 9, with excellent resistance to sulfidation. Yang et al. [57] developed a new cathode passivation treatment method for silver bonding wire containing a Pd solution. This method not only significantly improves the sulfidation resistance of silver bonding wire but also maintains its connectivity. Moreover, this treatment enhances the surface wettability of the wire on gold bond pads, suggesting that wetting modulation may be an effective approach for improving bond interface reliability. Wu et al. [58] studied a new type of flash-gold silver–palladium–gold alloy (AAPA) wire and found that the flash-gold film on the conductor’s surface could effectively improve the chlorine resistance and sulfide corrosion resistance of the wire.

### 3.2. Silver Bonding Wire Bonding Process

In addition to improving the performance of silver-based bonding wire via manufacturing process optimization, it is also necessary to optimize the bonding parameters so as to control the reliability of silver-based bonding wire well. Chen et al. [59] studied the effects of bonding parameters on the wire bonding properties of Ag-4Pd alloy. They point out that the bonding quality is greatly affected by a variety of process variables, among which the most important ones are bonding force, bonding time and bonding power. An increased bonding power can improve bonding performance, but there is an upper limit. It should be noted that excessive ultrasonic power will cause serious deformation of the bonded wire and reduce tensile properties. Fan et al. [60] investigated the influence of bonding parameters on the shape of the free air ball and the shape of the ball solder joint and wedge solder joint of Ag-0.8Au-0.3Pd-0.03Ru silver-based bonding wire. It is found that the diameter of FAB increases with an increase in bulb burning time and current when either the bulb burning current or the bulb burning time is constant. Additionally, when the bonding pressure is constant, the diameter of the welding spot pad of the bonding wire ball increases significantly with the increase in ultrasonic power. Cao et al. [61] studied the effects of EOF current and bonding power on the free air ball performance and bonding strength of Ag-10Au-3.6Pd alloy bonding wire. They found that EFO current and time would affect the shape of FAB. When EFO time was 0.80 ms and EFO current was 0.030a, FAB was a regular ball. For the studied bonding wire, the FAB diameter has a nonlinear relationship with EFO time at an EFO current of 0.030A, which can be expressed as a quadratic function. Furthermore, a too high or too low bonding power will lead to a reduction in bonding strength, so the bonding power should be within a certain range. Zhou et al. [62] studied the bonding properties of Ag-2.35Au-0.7Pd-0.2Pt-0.1Cu alloy (AAPPCA) wire with a diameter of 25 µm under different processing parameters. The effects of electrical breakdown (EFO) current and EFO time on the deformability of the free air ball (FAB) were studied using scanning electron microscopy (SEM), as well as the effects of ultrasonic power and bonding force on bonding characteristics. A set of bonding parameters suitable for AAPPCA wire was obtained. Under this set of bonding parameters, the tensile values of all samples in the destructive tensile test were greater than the standard values. Figure 10 shows a typical SEM image of the ball band after shearing. Intermetallic compounds (IMCs) are completely covered on the surface of the solder pad and have a regular morphology. The above tensile and shear test results indicate that all bonded samples have high bonding strength and high reliability. This indicates that, in addition to improving the bonding reliability of silver-based bonding wire via manufacturing processes, the adjustment of the bonding parameters to an optimal value can also effectively control the quality of wire bonding. The study of the bonding parameters of different bonding wires is an important aspect of the development of silver-based bonding wire.

When evaluating alternative materials, it is necessary to explore the manufacturing process and product quality of the new material bonding wire, as well as its optimal bonding parameter setting. It is helpful to establish a series of evaluation methods for judging the properties and reliability of new bonded materials. Lin et al. [63] comprehensively considered the loss of multiple responses and used the Taguchi method to study the feasibility and optimal bonding parameters of replacing gold wire with silver alloy wire in the bonding process of laser diode package wire. The process quality index can fully reflect the yield and quality level of the process. Yu et al. [64] proposed a fuzzy process quality evaluation model to evaluate the lead bonding process quality of integrated circuit packages.

## 4. Reliability of Silver Wire Bonding

Due to the low reliability of silver wire bonding, so far, silver wire has not been widely used in the electronic packaging field. The main failure causes of silver wire bonding include corrosion, electromigration, IMC failure, etc. The study of different failure modes of silver bond wire can make an overall judgment on the reliability of silver bond wire, and can also provide guidance for the improvement direction of silver bond wire and lay a good foundation for the large-scale promotion and application of silver wire.

An epoxy resin commonly used in the electronic packaging industry contains chlorine and sulfur ions. Vulcanization can create holes in the stress concentration and has a greater impact on the reliability of the wire than bonding. Chloride ions corrode the wire along the grain boundary, which leads to the change from ductile fracture to brittle fracture and results in a decrease in the reliability of the wire. In addition, chlorination can also lead to an increase in the resistance of the silver-based bonding wire, thus reducing the conductivity of the wire and causing circuit failure [58]. Therefore, it is necessary to add alloying elements to make silver alloys or perform special treatment on the surface of bonding wires to improve their resistance to vulcanization and chlorination. The addition of different alloy elements has different effects on the properties of bonding wires. Lin et al. [65] mention that adding 10 to 5 orders of magnitude of Al and Ti forms a dense oxide layer on the surface of silver wire via alloy precipitation, passivating the surface of the silver wire and reducing corrosion such as sulfurization and oxidation. In the metal protection industry, molecular film self-assembly technology has the advantages of a simple and controllable process, environmental protection and low pollution, good metal luster appearance, and low impedance change rate, which has attracted the attention of the wire-bonding industry [66]. Xu et al. [67] studied a silver bonding wire (OCA) with a surface coated with an organic film containing S and N, and analyzed its surface morphology, electrical properties and bonding performance via sulfurization corrosion resistance tests. Via comparative experiments using pure silver wire, it was found that the Ag_2_S generated on the surface of OCA wire was significantly reduced, with good resistance to sulfide corrosion, ensuring the metallic luster and low resistance performance of the silver wire and greatly improving the reliability of bonding. In addition, Au, Pb and other precious metal coatings can also play a protective role in reducing the corrosion of silver bonding wire. The newly developed surface flash-gold film technology can also play a role in improving the chlorine resistance and sulfide corrosion resistance of the wire, and it is expected to become a better method for solving the corrosion problem of silver wire.

The electromigration (EM) phenomenon was first proposed by the scientist Gerardin more than a hundred years ago. In the case of excessive current density, EM has a particularly serious impact on the reliability of electronic line connections, and is one of the most intractable problems in high-performance microelectronics components [68,69]. Due to the significant increase in power density and the development of ultra-fine pitch circuits for miniaturization, the problem caused by the EM phenomenon has become more prominent, and the main failures of modules are attributed to wear caused by EM at the wire interconnect [70,71]. Chen et al. [72] studied the microstructure evolution and electromigration failure mechanism of Ag alloy bond wire, and found that EM-induced failure of AG-4Pd bonding wire occurred in a phased manner. It is initially controlled by grain boundary diffusion and subsequently by surface diffusion due to the transformation of the microstructure. Guo et al. [73] studied the microstructure evolution of Ag-8Au-3Pd alloy wire during electromigration, and found that grain boundaries and surface are the main diffusion paths in the EM process. Surface diffusion plays a dominant role in the EM failure of alloy wire. Chuang et al. [74] measured the activation energy of electromigration of wire materials and confirmed that the surface diffusion of silver was the main driving force of electromigration. Wang et al. [75] studied the electromigration phenomenon of silver alloy wire bonding under the conditions of high temperature and electrical stress. They found that silver ions not only migrated on the surface of the wire, but also migrated inside the bonding wire, as shown in Figure 11. It is also suggested that both temperature and current density can promote the electromigration effect of silver ions. Sun et al. [76] proposed, via finite element simulation and experimental comparison, that electromigration was the main factor causing changes in the surface morphology of silver wires, and the contribution of thermal migration could be ignored. Figure 12 shows the finite element simulation results of silver wire current, temperature gradient, and total atomic flux divergence at a current density of 0.8 MA/cm^2^. The different behaviors of silver wire under different current densities are shown in Figure 13. The Joule heating caused by the supply current has a very significant accelerating effect on electromigration. When the current density is below 0.63 MA/cm^2^, the resistance and surface morphology remain almost unchanged. However, when the current density is high, the Joule heating in the silver wire leads to an increase in temperature, which can sinter the silver wire and reduce its resistance. Meanwhile, the high current density leads to significant EM phenomena and leads to the failure of the silver wire. They also proposed guidelines that, by providing a predetermined current density, the surface quality of silver thin wires could be effectively improved, with the hope of improving the electrical reliability and service life of silver thin wires in electronic devices. Mizushima et al. [77] proposed that ionic impurities could promote metal corrosion in microelectronic packaging. They demonstrated the mechanism of ion-induced corrosion via the destruction detection of corroded intermetallic compounds and atomic simulation. Both chloride and sulfate ions promote corrosion but attack different intermetallic compounds. As seen in Figure 14a, chloride ion (Cl^−^) corroded the Ag_2_Al layer. On the other hand, sulfate ion (SO_4_^2−^) corroded the Ag_4_Al layer (Figure 14b). The control of ion mobility in forming materials is an effective method for improving the reliability of pure silver wire packaging. Lin et al. [78] studied the effect of grain size on silver alloy wire by using a water drop experiment. It was found that the grain refinement of Ag-4Pd alloy wire significantly accelerated the ion migration of Ag-4Pd alloy wire. The experiment shows that the passivation film on the fine-crystal Ag-4Pd wire is easily broken down, resulting in a higher dissolution tendency of silver than that on the coarse-crystal Ag-4Pd wire.

EM failure is caused by the growth in grain size and the diffusion of surface atoms due to atomic transport. The thermal stability and mechanical properties of grain structure are improved by a large number of annealing twins. The anti-electromigration ability of annealing twins is also higher than that of other conventional grains. The electromigration failure of silver bonding wire can be improved by a proper annealing process to some extent. For surface atomic diffusion, this phenomenon can be improved by surface treatment, such as coating a layer of Au, Cu and other metals onto the surface or using chemical reactions to passivate the surface. Xiao et al. [79] mention that when Au coating is present, the Ag_2_O particle layer on the anode line becomes thinner, the dendrite contact time is delayed, and the current density during dendrite contact is also greatly reduced. These effects can inhibit electrolytic migration and contribute to the application of gold-plated silver bonding wires in electronic packaging. Chen et al. [80] mentioned that both a passivation film and addition of Pd could improve the corrosion resistance of silver-based bond wire, and the corrosion product AgCl generated by the corrosion of silver bond wire had a high ohmic resistance, which could inhibit further dissolution of the bond lines. In addition, because the alloy wire is cold-deformation processed, in the processing process, as the wire is processed more and more fine, the grain size will be gradually refined with the increase in the deformation of the wire. Therefore, the control of the grain-thinning trend in bonding wire processing is expected to be a method for reducing the electromigration of silver-based bonding wire.

Bonding reliability is closely related to the composition, morphology and properties of intermetallic compounds (IMC). The reliability of the wire bonding interface in electronic packaging mainly depends on the microstructure and thickness of the IMC formed between the chip liner and the bonding interface [81]. The evolution of the structure and properties of the IMC at the interface after the bonding of the silver-based bonding wire has been one of the most concerning issues in the industry [82]. The study of silver-based bonding wire intermetallic compounds (IMC) is the key to understanding the failure mechanism of silver bonded wire. Guo et al. [83] used backscattered electron (BSE) imaging technology to track the evolution process of the interface and found that IMC initially formed at the periphery of the bonding region. After short-term annealing treatment (175 °C 24 h), the hole in the center of the interface disappeared due to the dominant role of silver diffusion in the growth of IMC. You et al. [84] studied the formation and growth of IMC during bonding between 96% silver wire (96Ag-1Au-3Pd) with similar 2N gold wire properties and an Al pad. They conducted a high-temperature storage life test (HTST) and an unbiased high accelerated stress test with temperature and humidity (uHAST). They looked at the microstructure to determine its failure mechanism. It is found that the thickness of IMC becomes larger with an increase in temperature. Under uHAST conditions, Cl ion and water will cause oxidation and corrosion and finally form Al_2_O_3_, H_2_, which can cause cracks, and HCl, which can accelerate the cyclic reaction. Xi et al. [85] mentioned that bonding cracks caused by Ag–Al IMC corrosion in humid environments were the most common failure mode for silver wire. Preventing Cl and moisture around the interface between silver wire and aluminum sheet is crucial for improving the reliability of silver wire. Liao et al. [86] found that the combination of appropriate Pd coverage and good IMC coverage was the key to obtaining good reliability performance. During wire bonding, aluminum elements can be diffused at the interface to form Ag_2_Al, as well as Au–Al and PD–Al-related intermetallic compounds. The study of Hsu et al. [87] showed that a high current density would increase the growth rate of IMC, so that the formation rate of IMC on the anode side was faster than that on the cathode side. IMC at the bonding interface included Ag_3_Al and Ag_2_Al. In the high-temperature aging test, with the increase in temperature and the extension of aging time, the Al liner completely disappeared, Ag_3_Al began to transform into Ag_2_Al, and the final IMC was Ag_2_Al [88]. Cheng et al. [25] have found that Ag_2_Al is a hexagonal close-packed tissue alloy with high thermal stability and excellent reliability in mechanical adhesion and electrical contact. Au–Al and Pd–Al polyphase intermetallic compounds are formed during the bonding of Ag–Au–Pd with Al, as shown in Figure 15. The coexistence of these intermetallic compounds can inhibit interfacial corrosion and promote good bonding between the alloy wire and Al pad. The presence of Pd in the Ag_3_Al IMC layer may limit the growth rate of IMC. However, the growth of Ag_2_Al intermetallic compounds will lead to an increase in bonding interface resistance, resulting in a decrease in conductor conductivity [89]. Tseng et al. [43] found that the amount of Ag and the concentration of Pd in IMC were related to the diffusion rate of Ag in the Ag-Pd bond. Du et al. [90] studied silver alloy wire with different Ag contents and found that the silver content in silver alloy wire affected the electromigration rate, thus affecting the bonding reliability. Silver alloy wire with a high silver content has poor reliability. Huang et al. [91] mentioned that 3.5% Pd inhibited Ag–Al IMC, while Pd higher than 3.5% promoted the growth of IMC. Therefore, the content of precious-metal alloy components in the silver-based bonding wire is not the higher the better, and it needs to be limited to a certain range according to different circumstances.

In addition, temperature and humidity are also important factors affecting the reliability of silver wire bonding. High temperature and humidity may accelerate interfacial corrosion, oxidation, electromigration, and IMC growth behavior, thereby reducing the reliability of silver wire bonding. Du et al. [90] found via uHAST that silver alloy wires with higher Ag contents had poorer reliability. This may be due to the high temperature promoting the electromigration of silver alloy wire. You et al. [84] respectively studied the growth pattern of IMC under 2000 h of high-temperature-storage lifetime test (HTST) experimental conditions at 150 °C and 175 °C for silver wire. They found that the growth rate of IMC at 175 °C was higher than its growth rate at 150 °C. This indicates that IMC will become larger as the temperature increases. Liao et al. [7] found using high accelerated stress testing (HAST) that a 2N Ag wire exhibited uneven intermetallic compound growth and small voids at the interface between the Ag ball and the intermetallic compound layer, which was more pronounced at higher temperatures and longer aging times. The above research indicates that improving the stability of silver-based bonding wires under high temperature and humidity conditions is also an important research content in the future development of silver-based bonding wires.

It can be seen that the reliability of silver-based bonding wire is determined by many factors. The effects of corrosion, electromigration and IMC on bonding reliability are different. In addition to the study of a single factor, it is necessary to comprehensively study the influence of multiple factors on bond reliability, so as to obtain a more comprehensive and accurate bond reliability assessment. In addition, Jhon et al. [92] solved the technical challenge of sample unsealing in silver alloy wire packaging by using the same laser type as wafer laser grooving, and achieved very good unsealing results without damaging the surface of the silver alloy wire. This lays a good foundation for the research and evaluation of bonding reliability under various factors after the silver wire bonding package is formed.

## 5. Summary and Prospect

Wire bonding technology has been applied in the field of microelectronic packaging for more than 70 years. The continuous update of new materials and new technologies has promoted the rapid development of this field. Because of its good performance, gold wire is the earliest and most widely used in the field of wire bonding. However, as the price of gold continues to climb, it is imperative to develop a cost-effective and reliable bonding wire. Due to its good performance and low price, copper bonding wire has developed well in recent years and occupies a certain market share. However, the shortcomings of high hardness and low reliability cannot be effectively solved. Copper wire is not the best substitute for gold wire in high-end and military applications. Silver bonding wire, as a new bonding wire, performs well in terms of performance and shows great potential. In particular, coated silver wire is expected to become the most suitable substitute for gold wire due to its characteristics of oxidation resistance and corrosion resistance. At present, there are still many problems with silver bonding wire, such as electromigration failure, reliability shortage, etc., which need to be further studied and improved. In the future, silver-based bonding wire can be studied and optimized from the following aspects. First of all, the exploration of the ratio of different components in silver alloy wire and the addition of new elements, such as rare earth elements, can be considered to enhance the material’s mechanical and electrical properties. Secondly, it is also very meaningful to conduct more in-depth research on surface coating and surface treatment materials and processes. Surface treatment improves the oxidation resistance and corrosion resistance of the material and enhances the bonding reliability. Additionally, improvements in the preparation process of silver-based bonding wire are crucial. By controlling grain size during the wire preparation process using appropriate techniques, material mechanical performance can be enhanced and electromigration issues can be mitigated. Furthermore, it is necessary to sum up a set of bonding parameters suitable for silver-based bonding wire via experiments, ensuring higher reliability in silver wire bonding. Last but not least, further research into the growth and evolution patterns of intermetallic compounds (IMC) and the understanding of the mechanisms of IMC failure can lead to rational solutions for these challenges.

## Figures and Tables

**Figure 1 micromachines-14-02129-f001:**
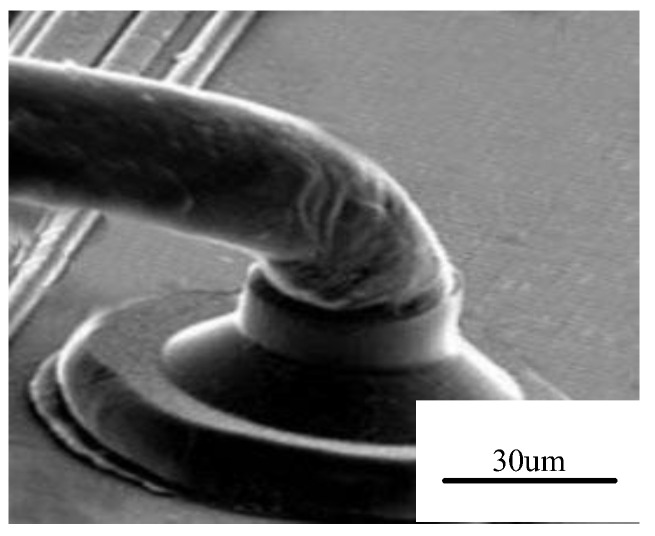
Solder joint deformation in high-temperature environments [13].

**Figure 2 micromachines-14-02129-f002:**
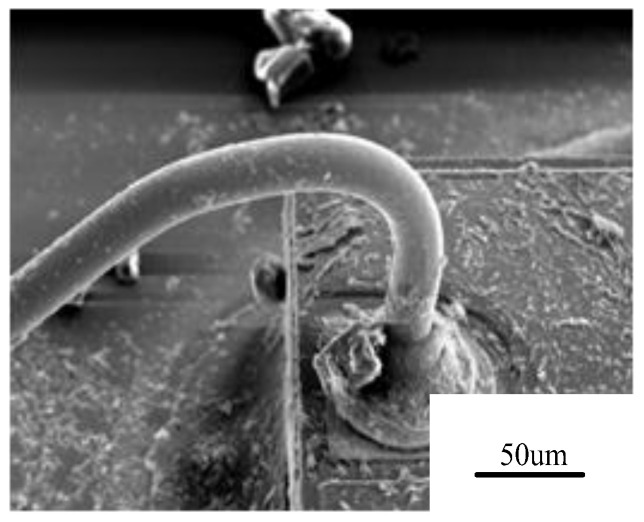
Ag bonding wire corrosion [13].

**Figure 3 micromachines-14-02129-f003:**
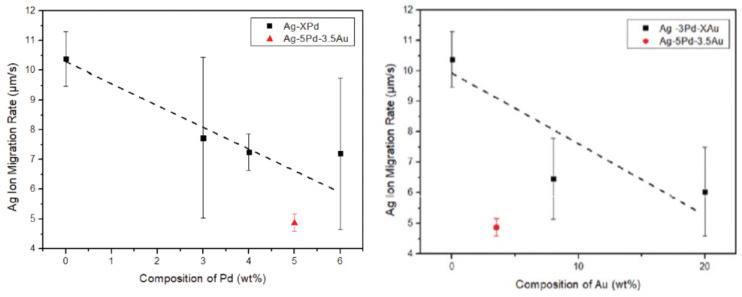
Ag ion migration rate of Ag bonding wires as a function of Pd and Au contents [22].

**Figure 4 micromachines-14-02129-f004:**
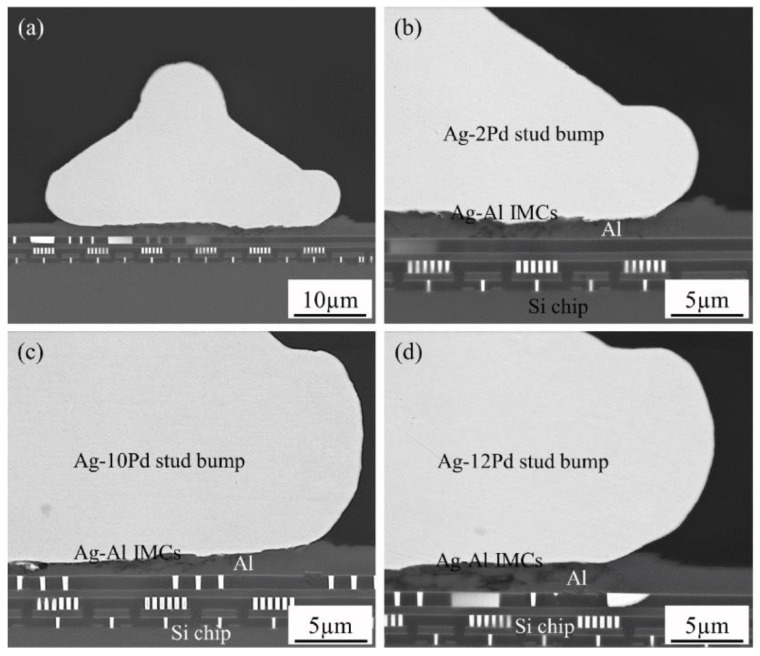
Morphology of the intermetallic compounds at the interfaces of the as-bonded Ag–Pd stud bumps on Al pads. (**a**) Standard Ag-2Pd stud bump bonded onto aluminum metallized Si chip, and magnified interfacial images of (**b**) Ag-2Pd, (**c**) Ag-10Pd and (**d**) Ag-12Pd [24].

**Figure 5 micromachines-14-02129-f005:**
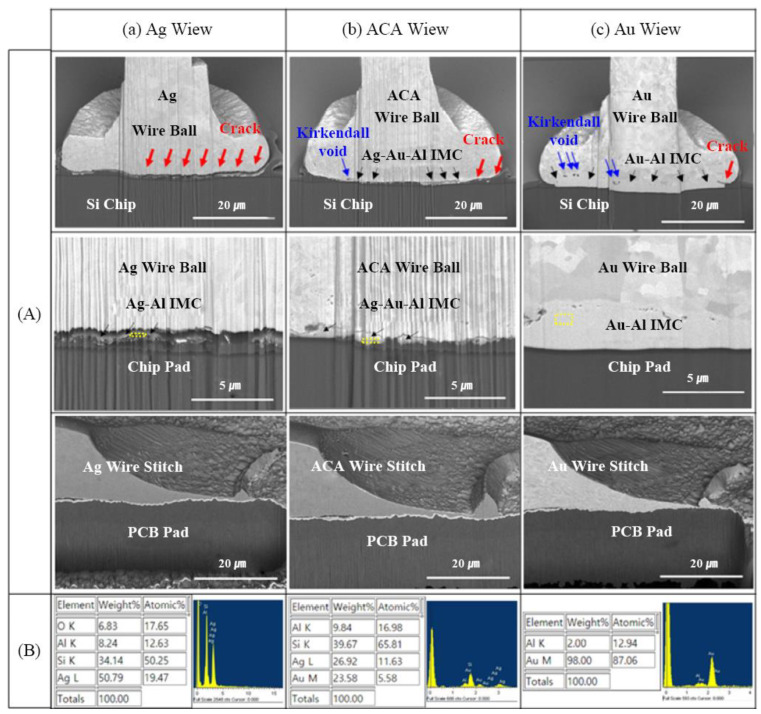
FIB cross-sectional (**A**) SEM micrographs and (**B**) EDS analysis results after reliability test; (**a**) Ag wire, (**b**) Au-coated Ag wire (ACA) and (**c**) Au wire [35].

**Figure 6 micromachines-14-02129-f006:**
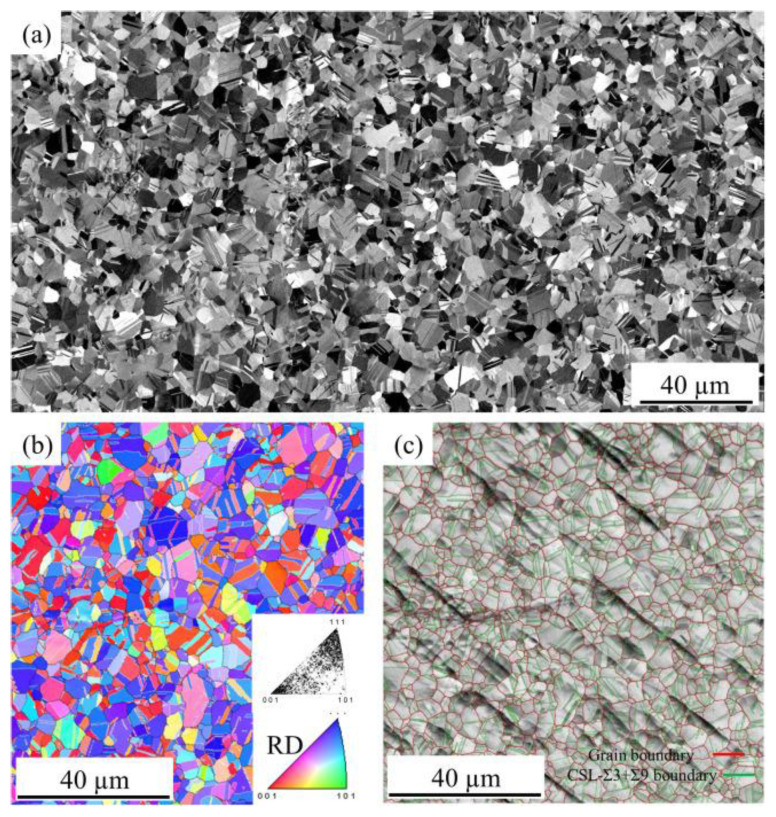
Microstructure of as-manufactured Ag-4Pd ribbon. (**a**) SEM image; (**b**) inverse pole figure map; (**c**) band contrast with grain boundary map [50].

**Figure 7 micromachines-14-02129-f007:**
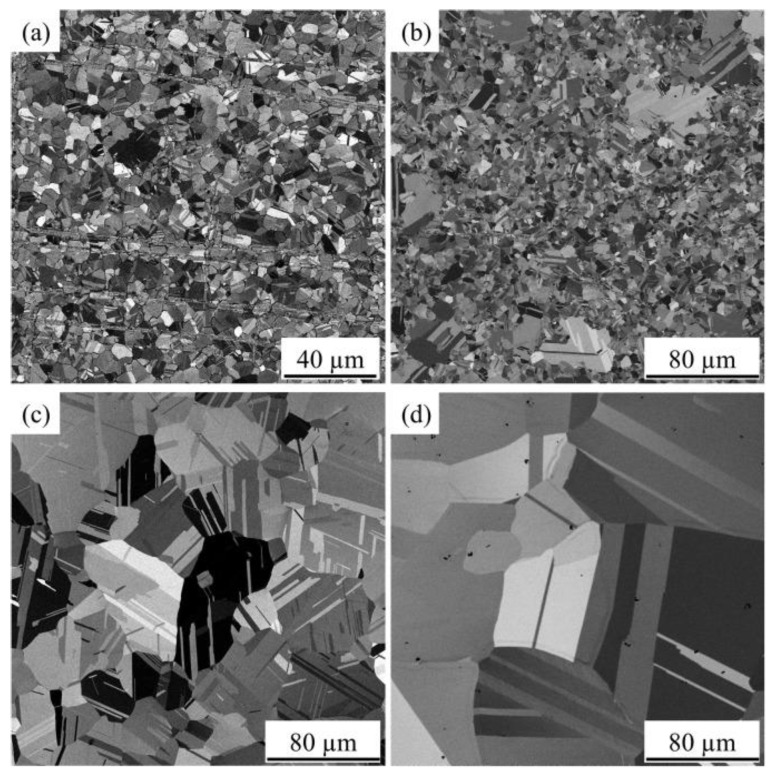
BSE images of Ag-4Pd ribbons after heat treatment at (**a**) 623 K, (**b**) 773 K, (**c**) 923 K, and (**d**) 1073 K for 100 h [50].

**Figure 8 micromachines-14-02129-f008:**
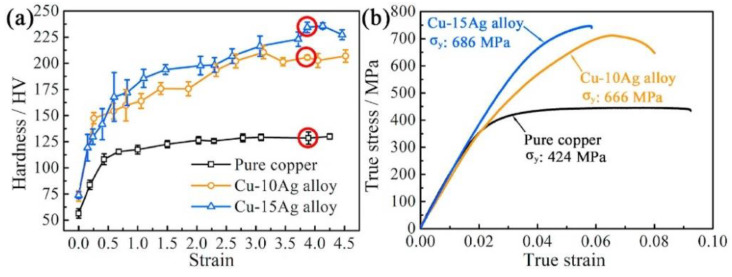
(**a**) Hardness and (**b**) stress–strain curve of pure copper, Cu–10Ag alloy and Cu–15Ag alloy after cold rolling; The red circle in (**a**) marks the hardness of the three materials at a rolling strain of 3.9 [51].

**Figure 9 micromachines-14-02129-f009:**
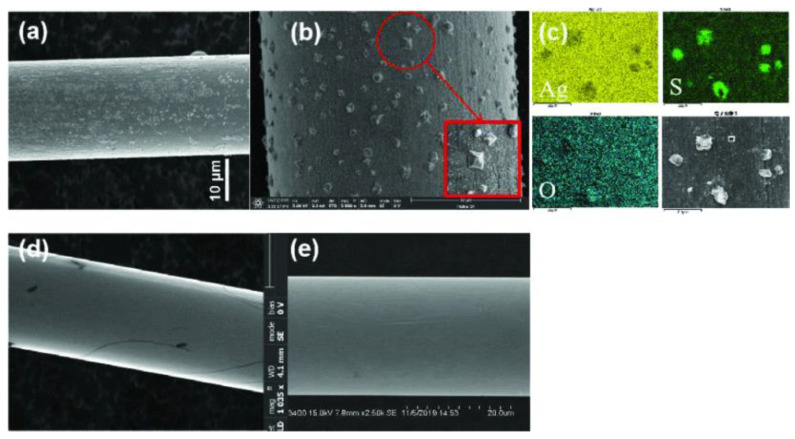
(**a**) Air sulfidation of untreated silver alloy bonding wire; (**b**) Surface particles of silver sulfide bonding wire; (**c**) Distribution of surface composition of sulfurized silver bonding wire; (**d**) Passivated silver alloy bonding wire; (**e**) Surface structure of passivated silver alloy bonding wire [56].

**Figure 10 micromachines-14-02129-f010:**
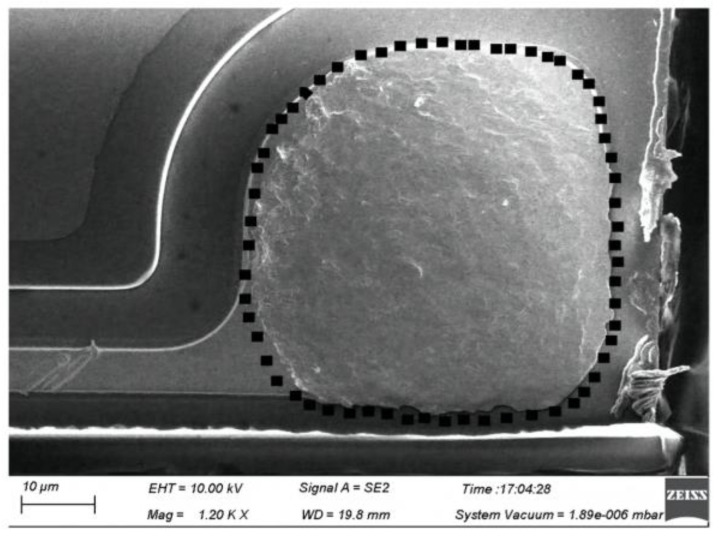
Typical ball bond morphology after ball shear test [62].

**Figure 11 micromachines-14-02129-f011:**
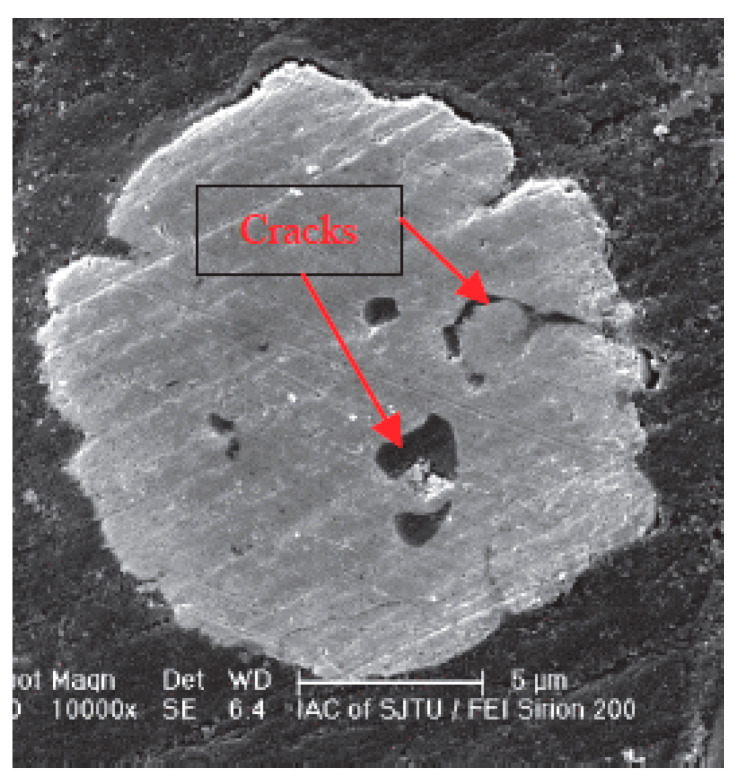
Cross-section of silver alloy wire after aging for 4 h [75].

**Figure 12 micromachines-14-02129-f012:**
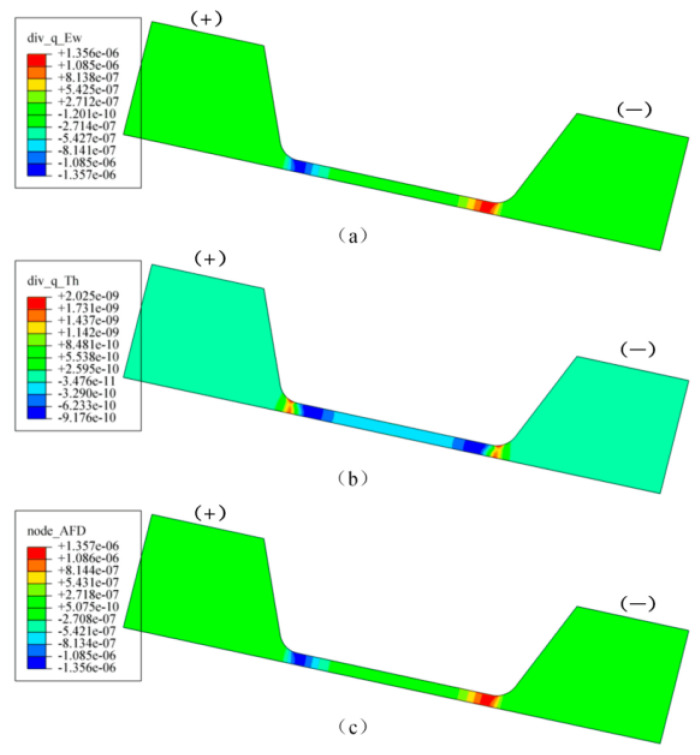
Distribution of atomic flux divergence by (**a**) current, (**b**) temperature gradient and (**c**) total atomic flux divergence (AFD) with current density of 0.8 MA/cm^2^ [76].

**Figure 13 micromachines-14-02129-f013:**
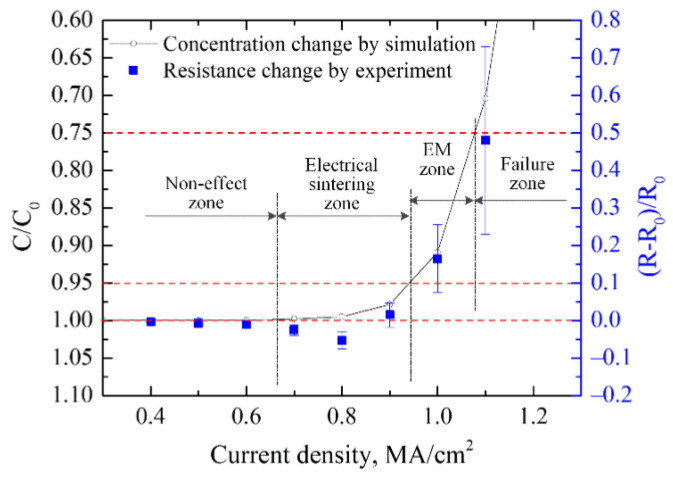
The different behavior of silver wire with current supply for 1 h at different densities [76].

**Figure 14 micromachines-14-02129-f014:**
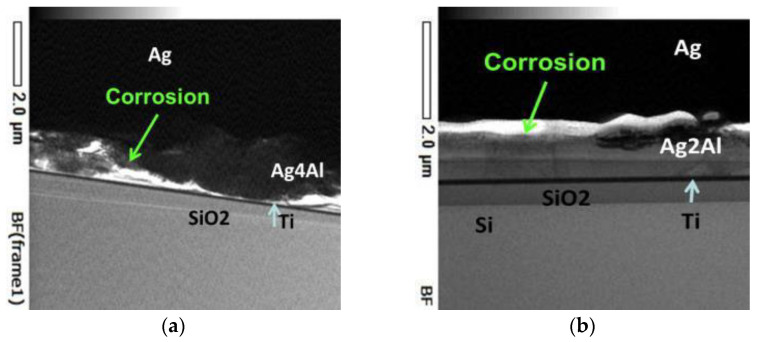
Cross-sectional TEM images of bonding interfaces: (**a**) pure Ag/CEL-A-2; (**b**) pure Ag/CEL-A-3 [77].

**Figure 15 micromachines-14-02129-f015:**
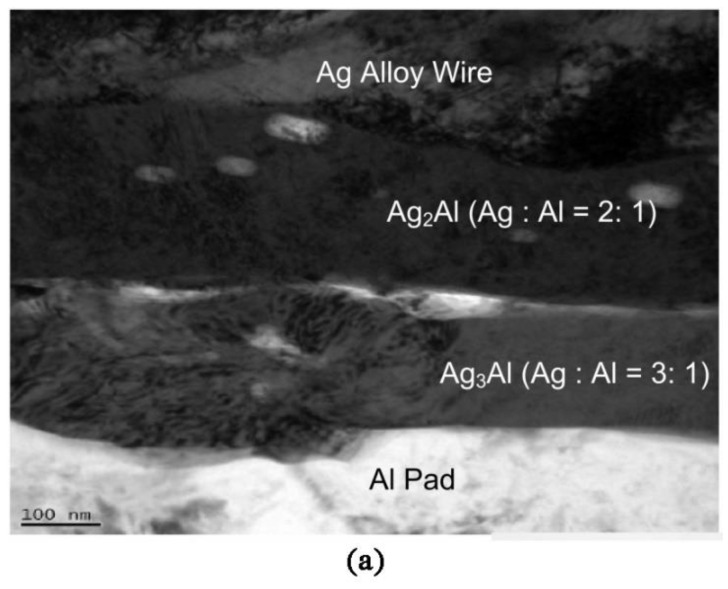
(**a**) The cross-sectional SEM image of the interfacial region prepared from as-bonded sample using FIB. (**b**) The EDS elemental mapping images of the corresponding area as shown in (**a**) [25].

**Table 1 micromachines-14-02129-t001:** Basic properties of pure Au, pure Cu and pure Ag [7,11].

Property	Units	Silver	Copper	Gold
Melting Point	°C	961.0	1083.0	1063.0
Thermal Expansion Coefficient	μm/m·K	19.0	16.7	14.2
Resistivity	×10^−9^ Ω·m	14.7	16.7	23.5
Thermal Conductivity	W/m·K	429.0	394.0	311.0
Electrical Conductivity	% IACS	108.4	103.1	73.4
Tensile Strength	MPa	125.0	209.0	103.0
Vickers Hardness	MPa	251.0	369.0	216.0
Yield Strength	MPa	35.0	33.3	40.0
Elastic Modulus	GPa	83.0	123.0	78.0
Brinell Hardness	HB	25.0	37.0	18.0
Metal Activity		Cu > Ag > Au		

## Data Availability

Not applicable.

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
