# Peer review of "A Review of Silver Wire Bonding Techniques"

_micromachines, 2023, doi:10.3390/mi14112129_

Round 1
Reviewer 1 Report
Comments and Suggestions for Authors
According the manuscript, three commons are discussed, including:
1. In part 2.1, please supplement some figures of pure silver wire bonding research.
2. In Fig. 4, whether the crystal orientation of EBSD characterization can be added?
3. As shown in “4.Reliability of silver wire bonding”, reliability includes many types, especially temperature. Please supplement the reviews of HTS, HAST, or TST, etc.
Reviewer 2 Report
Comments and Suggestions for Authors
The paper has a very comprehensive summary of the current state of the art of the Silver WBs. The readability of the paper is good as well. The review paper in the current state also has adequately referenced relevant work. Though more relevant work could be cited in the area of thermos-mechanical failures of the silver WBs. The topic is very relevant and has huge potential of widescale adoption of silver alloy WBs if certain challenges are solved.
Line 42 : “its” to “it’s”
Line 46 : “thermal conductivity and conductivity” to “Thermal conductivity and electrical conductivity”
Line 59 : “pure” to “Pure”
Line 89: what is silver wire ball solder joint ? this is confusing whether it is solder joint or silver wirebond on the pad side . The author is requested to make clarification.
Line 128 : The Ag2Al may require the 2 to be a subscript
Line 151 : ”wire” to “wires”
Line 290 : microwires to micro-wires
The author is requested to make “its” to “it’s” through out the paper
The authors are suggested to consider putting more review articles in the area of mechanical failures pertaining to silver wirebonds . these failures are also quite relevant to the current state of the art of silver wirebionds.
Comments on the Quality of English LanguageLine 42 : “its” to “it’s”
Line 46 : “thermal conductivity and conductivity” to “Thermal conductivity and electrical conductivity”
Line 59 : “pure” to “Pure”
Line 89: what is silver wire ball solder joint ? this is confusing whether it is solder joint or silver wirebond on the pad side . The author is requested to make clarification.
Line 128 : The Ag2Al may require the 2 to be a subscript
Line 151 : ”wire” to “wires”
Line 290 : microwires to micro-wires
The author is requested to make “its” to “it’s” through out the paper
